



# Application of linear minimum variance estimation to the multi-model ensemble of atmospheric radioactive Cs-137 with observations

Daisuke Goto[1], Yu Morino[1], Toshimasa Ohara[1], Tsuyoshi Thomas Sekiyama[2], Junya Uchida[3], and Teruyuki Nakajima[4]

[1]National Institute for Environmental Studies, Tsukuba, 305-8506, Japan
[2]Meteorological Research Institute, Tsukuba, 305-0052, Japan
[3]Atmosphere and Ocean Research Institute, University of the Tokyo, Kashiwa, 277-8568, Japan
[4]Earth Observation Research Center, Japan Aerospace Exploration Agency, Tsukuba, 305-8505, Japan

*Correspondence to*: Daisuke Goto (goto.daisuke@nies.go.jp)

**Abstract.** Great efforts have been made to simulate atmospheric pollutants, but their spatial and temporal distributions are still highly uncertain. Observations can measure their concentrations with high accuracy but cannot estimate their spatial distributions due to the sporadic locations of sites. Here, we propose an ensemble method by applying a linear minimum variance estimation (LMVE) between multi-model ensemble (MME) simulations and measurements to derive a more realistic distribution of atmospheric pollutants. The LMVE is a classical and basic version of data assimilation, although the estimation itself is still useful for obtaining the best estimates by combining simulations and observations without a large amount of computer resources, even for high-resolution models. In this study, we adopt the proposed methodology for atmospheric radioactive caesium (Cs-137) in atmospheric particles emitted from the Fukushima Daiichi Nuclear Power Station (FDNPS) accident in March 2011. The uniqueness of this approach includes (1) the availability of observed Cs-137 concentrations near the surface at approximately 100 sites, providing dense coverage over eastern Japan; (2) the simplicity of identifying the emission source of Cs-137 due to the point source of FDNPS; (3) the novelty of MME with the high-resolution model (3-km horizontal grid) over complex terrain in eastern Japan; and (4) the strong need to better estimate the Cs-137 distribution due to its inhalation exposure among residents in Japan. The ensemble size is six, including two atmospheric transport models (the Weather Research and Forecasting-Community Multi-scale Air Quality (WRF-CMAQ) model and non-hydrostatic icosahedral atmospheric model (NICAM)). The results showed that the MME-that estimated Cs-137 concentrations using all available sites had the lowest geometric mean bias (GMB) against the observations (GMB=1.53), the lowest uncertainties based on the root-mean-square error (RMSE) against the observations (RMSE=9.12 Bq m⁻³), the highest Pearson correlation coefficient (PCC) with the observations (PCC=0.59) and the highest fraction of data within a factor of 2 (FAC2) with the observations (FAC2=54%) compared to the single-model members, which provided higher biases (GMB=1.20-4.29), higher uncertainties (RMSE=19.2-51.2 Bq m⁻³), lower correlation coefficients (PCC=0.29-0.45) and lower precision (FAC2=10-29%). At the model grid, excluding the measurements, the MME-estimated Cs-137 concentration was estimated by a spatial interpolation of the variance used in the LMVE equation using the inverse distance



weights between the nearest two sites. To test this assumption, the available measurements were divided into two categories, i.e., learning and validation data; thus, the assumption for the spatial interpolation was found to guarantee a moderate PCC value (>0.4) within an approximate distance of 50 km. Extra sensitivity tests for several parameters, i.e., the site number and the weighting coefficients in the spatial interpolation, the time window in the LMVE and the ensemble size, were performed.

The most important assumption was that the ensemble size generated remarkably better results than the single-member model as it increased. Therefore, the proposed ensemble method, with a maximum ensemble size (six in this study), can be applicable for the best estimation of the Cs-137 distribution.

## 1 Introduction

Great efforts to simulate atmospheric pollutants have been carried out, but the spatial and temporal distributions of simulated
pollutants are still highly uncertain (e.g., Fuzzi et al., 2015). In contrast, observations are the most reliable method of monitoring the concentrations of atmospheric pollutants with high accuracy; however their spatial network is usually sporadic. Even if these observations densely cover the target area, they cannot reveal the pathway of pollutants from the source to the sink. To analyse the measurements and deeply understand their behaviours in the atmosphere, we need to improve atmospheric transport models as well as optimal interpolations using observations. To understand the model
performance, we have executed model intercomparison projects (MIPs) or multi-model ensembles (MMEs), which provide more reliable results than those by a single model for weather forecasting and climate prediction (e.g., Stensrud et al., 2000). To develop the optimal interpolation, we have also analysed the error and the variance between the simulations and observations to estimate more realistic distributions of the target materials (e.g., Rutherford, 1972; Talagrand, 1997; Robinchand and Ménard, 2014).
The MME technique is applied for weather forecasting (Stensrud et al., 2000; Gneiting and Raffery, 2005), climate projections (Knutti et al., 2010; Taylor et al., 2012), short-lived climate forcer assessments (Lamarque et al., 2013; Myhre et al., 2013), air quality forecasting (Solazzo et al., 2012; Sessions et al., 2015) and atmospheric dispersion predictions (Draxler et al., 2015; Sato et al., 2018). The members of the ensemble are widely spread for the use of various numerical models with perturbed initial conditions and various physical and chemical modules. The ensemble method is generally divided into three
types: pure average scheme (equal weighting), weighting scheme with all members and selected scheme with reduced members. The pure average method is a popular method for MME iclimate studies based on the concept of "one model, one vote" (Knutti et al., 2010; Weigel et al., 2010) or evidence of improvements in the concentrations of the pollutants in the air-quality and atmospheric dispersion simulations (McKeen et al., 2005; van Loon et al., 2007; Sessions et al., 2015; Kitayama et al., 2018). The weighting scheme, especially with a relatively smaller ensemble size, can be adopted to eliminate the
common biases and improve the ensemble results in the weather forecast (Krishnamurti et al., 1999), climate studies (Haughton et al., 2015), air-quality forecasts (Casanova and Ahrens, 2009) and atmospheric dispersion predictions (Nakajima et al., 2017; Sato et al., 2018). The selected scheme is used in the air-quality simulations (Solazzo et al., 2012;





Solazzo et al., 2013) and the atmospheric dispersion simulations (Riccio et al., 2012; Solazzo and Galmarini, 2015). The use of the non-pure average scheme, i.e., weighting and selected schemes, is increasing, and the technique has a useful tool for estimating the reliable results among MMEs.

In this study, the MME with the weighting scheme that minimizes the variance between the simulations and observations is

carried out to derive a more realistic distribution of radioactive caesium (Cs-137) at the surface. The Cs-137 in atmospheric particles was emitted from the Fukushima Daiichi Nuclear Power Station (FDNPS) accident in March 2011. Thus far, many atmospheric dispersion models have simulated Cs-137 aerosols (e.g., Chino et al., 2011; Morino et al., 2011; Stohl et al., 2012) and several MIPs were conducted (SCJ, 2014; Draxler et al., 2015; Sato et al., 2018). Under the MIPs, the MMEs provided reliable results in the assessed models (10 or more). However, ordinary modellers cannot easily carry out such

MMEs using only using their own models. For such situations, we propose a useful method for limiting ensemble size in MMEs by applying an analytical optimization to determine the weights for the ensemble. The optimization is based on a linear minimum variance estimation (LMVE), which is a classical and basic type of data assimilation similar to the Kalman filter (e.g., Talagrand, 1997; Kanlay, 2003).

This approach is unique from other MMEs of other species based on the following four factors: (1) The observed Cs-137

concentration near the surface is available at approximately 100 sites, providing dense coverage of eastern Japan (Tsuruta et al., 2014; Oura et al., 2015). Since plumes including Cs-137 particles are transported and diffused very heterogeneously, dense measurements are essential to capture such plumes. (2) The spatial distribution of Cs-137 distribution is captured relatively easily since Cs-137 is emitted from the point source. For example, the PM2.5 distribution is rather difficult to capture in the atmosphere since it is emitted from complex sources and formed from various chemical reactions. (3) MME

studies to identify a simple tracer with a high-resolution (3 km horizontal grid) model over complex terrain, such as Fukushima, are still limited. (4) It is very important for people to properly estimate the spatial and temporal distributions of Cs-137 emitted from the FDNPS. The better estimation of the Cs-137 distribution greatly helps us to understand the impacts of inhalation exposure on residents in Japan.

In a previous study, Nakajima et al. (2017), which comprises the basis of our proposed method, was applied using multi-

models, including the Weather Research and Forecasting-Community Multi-scale Air Quality (WRF-CMAQ) model (Morino et al., 2013) and non-hydrostatic icosahedral atmospheric model (NICAM) (Goto et al., 2018), to derive a better Cs-137 distribution. However, the estimation is still uncertain, and its ensemble results were not greatly improved, which was mainly because the results of the original model were still highly uncertain. In addition, Nakajima et al. (2017) did not discuss the availability of the use of LMVE for more than three members and the uncertainty of the relevant parameters in

LMVE. After the study of Nakajima et al. (2017), both models were further developed by using finer horizontal resolution (3 km) grid and by nudging a new meteorological field provided by Sekiyama et al. (2017) with higher accuracy. In this study, the available results were increased via the use of six members, including two atmospheric transport models and two or four sensitivity experiments. This proposed method using more than two members promises to be applicable for MIPs as a new



ensemble method. Furthermore, the estimated Cs-137 concentrations will be used for the estimation of inhalation exposure of Cs-137 emitted from FDNPS in March 2011 (Takagi et al., to be submitted).

Section 2 gives a description of two models, WRF-CMAQ and NICAM, including a design of the sensitivity experiments, an explanation of the ensemble method using LMVE, the used measurement datasets with designs to test several assumptions in

the proposed ensemble method, and statistical metrics for model evaluation. Section 3.1 shows the estimated Cs-137 concentrations and their comparison with the single-model results. Section 3.2 shows the tests used for the assumption of spatial interpolation in the LMVE equation using the distance between the nearest two sites, as shown in Section 2.2, by separating the measurement into learning and validation sites. In the proposed method, several parameters are assumed: the size, number and weighting values in the interpolation, the time window in the LMVE and the ensemble size. These

parameters are investigated and discussed in Section 4. Section 5 shows a conclusion and the implication of this study.

## 2 Methods

### 2.1 Description of the two atmospheric transport models

The ensemble size is six, and it includes two different atmospheric transport models, the WRF version 3.1 (Skamarock et al.,

2008) coupled with the CMAQ version 4.6 (Byun and Schere, 2006) and the NICAM (Tomita and Satoh, 2004; Satoh et al., 2008; Satoh et al., 2014) coupled with the spectral radiation-transport model for aerosol species (SPRINTARS; Takemura et al., 2005; Goto et al., 2011). According to the rule of Nakajima et al. (2017), these models are hereafter referred to as the W-model and N-model, respectively. The W-model analyses atmospheric processes, such as transport, diffusion and deposition of particles, but it was modified for radioactive particle use, such as Cs-137 emitted from the FDNPS accident in the target

area in Japan (Morino et al., 2013). The basic experimental design in this study is common, such as in Morino et al. (2013). The N-model is a seamlessly multi-scaled model for air pollutants (Goto et al., 2018) on a global scale with a quasi-uniform grid (Suzuki et al., 2008; Dai et al., 2014), a semi-regional scale with a stretched grid (Goto et al., 2015; Goto et al., 2019), and a perfect regional scale with a diamond grid system (Uchida et al., 2017; Nakajima et al., 2017). The N-model also considers the atmospheric processes of particles and focuses on the target area in this study. The basic experimental design in

this study is generally common to Nakajima et al. (2017). Both the W-model and N-model participate in the international MIP for Cs-137 emitted from the FDNPS accident (Sato et al., 2018). The experimental design among the models was harmonized as best as possible, that is, all experiments were carried out by using the same emission inventory in Katata et al. (2015) and nudging the meteorological fields using the operational model for regional weather forecasting around Japan (the non-hydrostatic model, named NHM; Saito et al., 2006) coupled with the LETKF (NHM-LETKF) from Sekiyama et al.

(2017), with almost a 3-km grid resolution. Using the W-model and N-model, six experiments were conducted, as shown in Table 1. The W-model was executed in four experiments by considering differences in the meteorological fields used as nudging data, the wet deposition process for Cs-137, and emission scenarios of Cs-137 from Terada et al. (2012). The N-



model was performed in two experiments by considering only the difference in the meteorological fields. The hourly Cs-137 concentrations simulated in all the experiments in the lowest layer are linearly interpolated to a 1 km × 1 km grid cell (3$^{rd}$ mesh grid filled for people's living areas to indicate inhalation exposure in Japan in Takagi et al. (to be submitted)) for the ensemble process. The target region is eastern Japan as shown in Figure 1. The target period is from 11 March to 24 March

2011 (Japan Standard Time; JST).

## 2.2 Ensemble method

One of the optimization methods for the simulated Cs-137 concentration is a multi-model ensemble. When the simulated concentration in model $i$ represents $C_i$ and the number of models is two, the ensemble concentration $C_{ens}$ can be expressed as

$$C_{ens} = a_1 C_1 + a_2 C_2, \quad (1)$$

where $a_i$ ($i=1,2$) is a weighted coefficient for $C_i$ ($i=1,2$). According to the idea of LMVE used in Nakajima et al. (2017), this study also defines the weighted coefficient as:

$$a_i = (1/\sigma_i^2)/(1/\sigma_1^2 + 1/\sigma_2^2), \quad (2)$$

where $\sigma^2$ represents a variance between the simulated Cs-137 ($C_{sim}$) and observed Cs-137 ($C_{obs}$).

$\sigma^2$ is defined as

$$\sigma^2 = \sum_{dS,dt} (C_{sim} - C_{obs})^2, \quad (3)$$

where $dS$ is a spatial window, i.e., a specific domain, and $dt$ is a time window, i.e., a specific period. This formulation is classical and widely used for various subjects, such as data assimilation (e.g., Rutherford, 1972; Talagrand, 1997; Kalnay, 2003), and it can be applicable for MMEs (Potempski and Galmarini, 2009). This method leads to the minimization of the

difference between the simulation and observation results without a large amount of computer resources.

Different from the previous study of Nakajima et al. (2017), the number of ensemble members is not only two but also more than two. In this case, Eqs. (1) and (2) are generalized as follows:

$$C_{ens} = \sum_{i=1}^{N} a_i C_i, \quad (4)$$

$$a_i = (1/\sigma_i^2)/\sum_{j=1}^{N} (1/\sigma_j^2), \quad (5)$$

where $N$ is the number of ensemble members. Eq. (5) represents the non-biased conditions and the weight is satisfied as follows:

$$\sum_{j=1}^{N} a_j = 1, \quad (6)$$

In this study, the term $C_i$ in Eqs (3) and (4) is the logarithmic scale of Cs-137 concentrations because the Cs-137 concentration is very heterologous and ranges from 0.1 Bq m$^{-3}$ to 100000 Bq m$^{-3}$ (Tsuruta et al., 2014). When the observed

Cs-137 concentration is less than the detection limit, it is assumed to be 0.01 Bq m$^{-3}$ in Eq. (3).

In Eq (3), $dt$ is set to 1 hour, which is an assumption and tested by setting $dt$ to 3-48 hours (discussed in Section 4.1); and $dS$ is set to one grid, i.e., 1 km by 1 km, where the observation site is located. In grids where an observation site is missing, $\sigma^2$ is





geostatistically calculated by interpolating two $\sigma^2$ values at the nearest two observation sites (grids). In this study, $C_{ens}$ is not directly interpolated because it varies abruptly in space and time. The interpolation represents the spatial linearity for $\sigma^2$ between the nearest two observation grids, where the weighted coefficient depends on the inverse distance between the target grid and the two nearest observation grids, which is known as inverse distance weighting (IDW) and shown in Table 2 as

LIP 1. The dependence on distance has been implemented in previous studies (e.g., Rutherford, 1972; Hollingsworth and Lönnberg, 1986). To confirm this assumption, an additional four IDW methods were also carried out as shown in Table 2 (discussed in Section 4.2). One of them is the nearest neighbours, which is a popular method of interpolating the concentrations; however, the spatial pattern of the $\sigma^2$ value is not smoothly distributed. In two of the sensitivity tests, the dependence of the $\sigma^2$ value on distance is stronger than that of the other method based on an inverse square-distance to the

interpolation. The ensemble size, $N$, is set to six, which is also tested by changing the number from six to two, three, four and five (discussed in Section 4.3). Hereafter, the proposed method is referred to as the 'LMVE ensemble method'.

### 2.3 Observation data

The hourly measured Cs-137 concentrations at the surface are directly estimated by using the aerosol sampling tapes of the

national suspended particulate matter (SPM) network (Tsuruta et al., 2014; Tsuruta et al., 2018). There are almost 400 SPM sites in eastern Japan, but now 101 sites have available data for Cs-137 (Oura et al., 2015). In this study, the measured Cs-137 data at 100 sites (one site, Futaba, was eliminated because the site is located in the same model grid as FDNPS, which is known as the change-of-support problem (Gotway and Young, 2002)) are used for the ensemble. In addition, at an extra site, Tokai (140.59°E, 36.45°N), which covers the missing area of Oura et al. (2015), daily measured Cs-137 concentrations

(Furuta et al., 2011) are used. All 101 sites used in this study are plotted using four different colours in Figure 1. All measurements shown in Figure 1 (all colour sites) are used as learning data in the LMVE ensemble method as a control experiment (CTL).

The colours represent the sites used to learn or validate the data in the extra ensemble estimation to examine the effect of the spatial interpolation for the $\sigma^2$ value on the MME concentrations. In the test experiments (SEN1, SEN2 and SEN3), some of

the datasets are used as learning data and others are used as validation data, as shown in Table 3. The validation data are selected by choosing one of the sites as the representative site in the specific domain. When the specific domain is defined by a lattice of 0.125° by 0.125°, the sites in yellow are used as validation data for the experiments (SEN3). When the specific domain is defined as a lattice of 0.25° by 0.25°, the sites in yellow and green are used as validation data in the experiments (SEN2 and SEN3, respectively). When the specific domain is defined as a lattice of 0.5° by 0.5°, the sites in yellow, green

and blue are used as validation data for the experiments (SEN1, SEN2 and SEN3). This means that in the SEN2 experiment, the 56 sites in blue and green can be used as validation data, whereas the 45 sites in red and yellow are used as learning data.





Since the area where Cs-137 is emitted from is the point source of FDNPS, a method used to randomly choose the learning/validation sites is not applied for accurately evaluating the spatial interpolation of the ensemble method.

**2.4 Statistic metrics for the MME evaluation**

The model evaluation should be carried out using multiple statistical metrics (e.g., Chang and Hanna, 2004). In this study, we introduce the geometric mean bias (GMB), root-mean-square-error (RMSE) using the geometric variance (GV), Pearson correlation coefficient (PCC), and the fraction of data within a factor of two of observations (FAC2):

$$GMB = exp\left(\overline{\log C_{obs}} - \overline{\log C_{sim}}\right) \tag{7}$$

$$RMSE = \ln(GV^2) \tag{8}$$

$$GV = exp\left\{\overline{(\log C_{obs} - \log C_{sim})^2}\right\} \tag{9}$$

$$PCC = \frac{\sum(\log C_{obs} - \log \overline{C_{obs}})(\log C_{sim} - \log \overline{C_{sim}})}{\sqrt{\sum(\log C_{obs} - \log \overline{C_{obs}})^2 \sum(\log C_{sim} - \log \overline{C_{sim}})^2}} \tag{10}$$

$$FAC2 = fraction\ of\ data\ that\ satisfy; 0.5 \leq \frac{\log C_{sim}}{\log C_{obs}} \leq 2.0 \tag{11}$$

Because Cs-137 concentrations vary by several orders (e.g., Tsuruta et al., 2014), GMB and GV are preferred over the fractional bias method because they are suitable for extreme high and low values (Chang and Hanna, 2004). The RMSE is an

indicator of uncertainty. In this study, the PCC is estimated using logarithmically scaled Cs-137 concentrations. Because the logarithmic Cs-137 is undefined for zero values, a minimum threshold of logarithmic Cs-137 is set to the lower threshold in the measurement, i.e., 0.01 Bq m$^{-3}$. The PCC using logarithmically scaled values is actually not a robust measure because the lower limitation is sensitive to the results. However, the PCC using logarithmically scaled values is a useful indicator because it becomes applicable even for highly skewed distributions; therefore, we also use the PCC for the model evaluation.

FAC2 becomes the most flexible metric for evaluating the model results even if its probability distribution frequency is highly skewed, and it can be an indicator of precision. These statistical metrics are calculated for the case in which the observed Cs-137 exceeds the detection limit. The total sampling number is 7056 for all available sites and times in the CTL experiment, whereas it is 1865 (minimum value) for the SEN1 experiment. Because the sampling number is adequately large, a direct comparison of these statistical metrics among the different experiments can be performed.



## 3 Results

### 3.1 LMVE ensemble method using all available observations

Cs-137 simulated by parts of the ensemble six members, i.e., W1 and N1, is evaluated under an MIP (Sato et al., 2018). The performances of these two members are moderate among the MIP-participating models. Here, all of the ensemble members

and the ensemble results are compared with the measurements of the surface Cs-137 concentrations. Figure 2 shows the temporal variation in both simulated and observed Cs-137 at the sites near FDNPS and in the Kantou region. Generally, the ensemble results at these sites are the closest to the observations compared to the results of the single-member model. For example, at Naraha (Figure 2(a)), which is the closest site to FDNPS, the 1st and 2nd largest peaks in the observed Cs-137 are noteworthy during 15-17 March. The results of the ensemble members are largely dispersed, so the ensemble results become

very close to the observations. In the other peaks, such as those on 16 March and 20-21 March at Furukawa (Figures 2(c)), the ensemble results are almost completely matched with the observations. However, the ensemble results are not close to the observations, when all the results of the ensemble members are underestimated compared to the observation, as is the case on 13 March at Haramachi (Figure 2(b)). In this case, the ensemble result has a peak on 12 March, which is earlier than the timing captured by the observation. In contrast, on 12 March at Naraha, some of the members have a peak in the

simulated Cs-137; thus, the ensemble result also has a small peak, whereas the observation does not have such a peak. These cases are also shown on 20 March at Kawagoe (Figure 2(d)), which can be explained as follows: when the Cs-137 simulated by all members is underestimated compared to the observations, the variance in Cs-137 between the observations and the simulations, as defined in Eq. (3), must be too large and, thus, the weighted coefficient of the members, as defined in Eq. (5), becomes very small. Because the cross terms of the Cs-137 concentration and the weighted coefficient are small, the Cs-137

concentrations estimated by the ensemble must be underestimated. In contrast, when Cs-137 simulated by some members is overestimated compared to the observations, the weighted coefficient becomes very small. However, because the cross terms of the Cs-137 concentration and the weighted coefficient are not small, the Cs-137 concentrations estimated by the ensemble are overestimated, which represents one of the disadvantages of the LMVE ensemble method and prevents it from obtaining more accurate ensemble results relative to the observations.

Figure 3 shows scatterplots for the observed and simulated Cs-137 at all sites using the ensemble results (a), the results of each member (b-g) and the median results (h) among the ensemble members. The statistical metrics are listed in Figure 4, including the bias (GMB), uncertainty (RMSE), correlation (PCC) and FAC2. The perfect model presents values of GMB=1 ($C_{sim}=C_{obs}$), RMSE=0 Bq m$^{-3}$, PCC=1 and FAC2=1 (100%). The observation values at all 101 sites are used in the LMVE ensemble method as learning data. Compared with the results of the other ensemble members, the ensemble result is the

closest to the observations, with GMB=1.53, RMSE=$10^{1.709}$ (=9.42) Bq m$^{-3}$ (i.e., the lowest uncertainty), PCC=0.59 and FAC2=54%. The ensemble members produce slightly overestimated results ranging from GMB=1.21 to GMB=2.54 (for W1, W2, W3 and W4) and considerably overestimated results ranging from GMB=4.20 to GMB=4.30 (for N1 and N2), and they




have low-to-moderate correlations ranging from PCC=0.29 to PCC=0.45 and uncertainties ranging from $10^{1.283}$ (=19.2) Bq m$^{-3}$ to $10^{1.709}$ (=51.2) Bq m$^{-3}$. The median result using all ensemble members has a moderate correlation of PCC=0.42 but a high GMB of 4.39 and high RMSE of $10^{1.717}$ (=50.7) Bq m$^{-3}$. The median and average value using many members is generally close to the best estimate compared to the original models (e.g., Draxler et al., 2015), although in this study, the

median among the six members does not provide the best results.

### 3.2 LMVE ensemble method using limited observations

In Section 3.1, the results of the CTL are shown, and in this section, the sensitivity tests (SEN1, SEN2 and SEN3 as shown in Section 2.3) for the separation of learning and validation sites are conducted. The test results are evaluated at the sites that are independent (used as validation data) from the other sites (used as learning data) in the LMVE ensemble method (Table

3). At the independent sites, the temporal variations in the simulated Cs-137 are compared at the four sites near FDNPS and in the Kantou region (Figure 5). The sensitivity depends on the location; the results in SEN3 are far from the observations at Fukushima, as shown in Figure 5(a) and 5(c), whereas those of all the sensitivity tests (SEN1, SEN2 and SEN3) are generally close to the observations at the sites in the Kantou region (Figures 5(c) and 5(d)). This suggests that the interpolation of variance (and thus the Cs-137 concentrations) near the FDNPS is sometimes not applicable, which is

probably because the plume of high-density Cs-137 near the FDNPS is very narrow and strongly depends on local winds (Nakajima et al., 2017). The wind, especially low wind speeds, tends to influence the results at the observation sites (Weil et al., 1992).

Figure 6 summarizes the statistical metrics among the sensitivity tests. This figure indicates that the GMB and RMSE values are larger and the PCC and FAC2 values are smaller as the distance between the learning and validation sites increases. The

figure also shows that the differences in these metrics between the results using all and independent sites are small and thus clearly show the success of the linear interpolation of variance between the simulation and the observation in the LMVE ensemble method. These results are consistent with the results shown in the Kantou region of Figures 5(c) and 5(d), indicating that the largely spread plumes are generally reproduced by the ensemble method. As shown in Figure 6, the relationship between the two axes, i.e., distance vs PCC, can be fitted as a linear line with a slope of 0.43 in both results

using all/independent sites. The slope indicates that the PCC decreases by approximately 0.04 when the distance from the learning site to the validation site increases by 0.1°. According to the approximate line, the distance is calculated to be 0.53° when all sites are used and 0.42° when the independent sites are used to obtain moderate correlation (PCC>0.4). Therefore, the proposed interpolation is generally applicable for the best estimation within a distance of 0.4°-0.5°, i.e., approximately 50 km, from the observation site.





## 4 Discussion

This section discusses the uncertainties caused by several assumptions in the LMVE ensemble method. As described in Section 2.2, the spatial interpolation of variance defined in Eq. (3) assumes IDW. The selection of the sites is also uncertain and is investigated in Section 4.1. The time window used in Eq. (3) is assumed to be 1 hour, which is discussed in Section
4.2. The ensemble size defined in Eqs. (4) and (5) is also discussed in Section 4.3. In Section 4.4, the spatial distribution of the Cs-137 surface concentrations estimated by the LMVE ensemble method is shown as an example for estimating the impact of Cs-137 on inhalation exposure of residents in Japan.

### 4.1 Sensitivity of spatial interpolation

The spatial interpolation of variance defined in Eq. (3) adopts IDW using two of the nearest sites as denoted by LIP1 in
Table 2. Here, four extra methods in Table 2 are used for testing SEN1, SEN2 and SEN3, as shown in Section 3.2. Figure 7 illustrates the statistical metrics for 15 tests using the validation sites, which are not used in the LMVE ensemble method as learning data. For SEN1, all metrics in the five interpolation methods are estimated as 1.60±0.00 for GMB, $10^{0.95±0.00}$ Bq m$^{-3}$ for RMSE , 0.63±0.00 for PCC and 52±0.0% for FAC2 by using 1865 data asmples. The differences among the interpolation methods are close to zero. In SEN2 and SEN3, however, the differences among the 5 interpolation methods, especially
between the nearest neighbours (named Nearest in Table 2) and the others, become slightly larger than the others. The largest difference is observed between *Nearest* and the others at 0.06 for GMB, $10^{0.04}$ Bq m$^{-3}$ for RMSE, 0.03 for PCC and 1% for FAC2. Judging from the slight differences and the simplest method, we use the LIP1 method (using the two nearest sites and the weighting coefficient of the inverse distance) in the standard experiment. It should be noted that other interpolations were considered in the discussion, although they provided much worse results than those shown in Figure 7
(not shown). The method that used the covariance between the observations at the nearest observation site and the simulation at the target grid was not applicable to this study because the covariance values are generally negative or close to zero at most grids due to the heterogenous distribution of Cs-137. Another method that uses all sites (not two or three grids) for the interpolation was also not applicable to this study because the influence of the ensemble coefficients at the grids far from the target grid cannot be ignored.

### 4.2 Sensitivity of the time window

To investigate the sensitivity of the time window in Eq. (3), the temporal variations in Cs-137 simulated by the ensemble methods are shown in Figure 8 using various time windows ranging from 1 hour to 49 hours (not all results are shown in Figure 8). The difference in the estimated Cs-137 concentrations is generally small but sometimes very large. In Figure 8 (a),
for example, at Naraha on 20 March, the observed peak is sharp, whereas the sharpness of the estimated peaks depends on





the time window values. As the time window increases, the sharpness of the peak becomes weak, i.e., the peak is broadly distributed. At Kawagoe (Figure 8(d)) on 20-22 March, the estimated Cs-137 concentrations using the longer time window are far from the observations and estimations using the shorter time window. Such situations are found at the other sites and during other periods (not shown). This also indicates that the peak in Cs-137 is very sharp temporally and spatially, so the

time window must be shortened. The dependency of the time window on the results is investigated using the statistical metrics at all sites used in the LMVE ensemble method, as shown in Figure 9. The dependency of the time window on the GMB, RMSE and PCC was found to be very weak while that on FAC2 was strong; moreover a shorter time window tends to provide a higher PCC and lower GMD and RMSE values. Therefore, the time window in the standard experiment is set to the shortest time, i.e., 1 hour.

### 4.3 Sensitivity of the ensemble size

The previous study of Nakajima et al. (2017) used only two members for the LMVE ensemble method, and the ensemble results were better than the original results for each member, but the difference in the PCC was very small (0.03-0.05; Table 1 in Nakajima et al., 2017). Therefore, this study increases the number of LMVE ensembles to six members and investigates

the sensitivity of the ensemble size to the results. Figure 10 shows the relationship between the ensemble size and the statistical metrics (GMB, RMSE, PCC and FAC2). The results clearly shows that as the ensemble size increases, the GMB and RMSE decrease and the PCC and FAC2 increase. Using two members, the average GMB is calculated to be 1.95, which is smaller than that using a single member by 0.76; the average RMSE is calculated to be $10^{1.208}$ (=16.2) Bq m$^{-3}$, which is smaller than that using a single member by 14.3 Bq m$^{-3}$; the average PCC is calculated to be 0.46, which is larger than that

using a single member by 0.08; and the average FAC2 is calculated to be 34%, which is larger than that using a single member by 14%. Using more than two members, the PCC is calculated to be more than 0.4, i.e., a moderate correlation. Using five members, the average GMB is calculated to be 1.56, which is larger than the value of 1.53 using six members; the average RMSE is calculated to be $10^{0.984}$ (=9.63) Bq m$^{-3}$ but larger than the value of $10^{0.960}$ (=9.12) Bq m$^{-3}$ using six members; the average PCC is calculated to be 0.57, which is smaller than the value of 0.59 using six members, and the

average FAC2 is calculated to be 51% but smaller than the value of 54% using six members. For the median and average values using six members, the PCC is calculated to be 0.42 and 0.46, respectively, which are similar to the ensemble results using two members. In contrast, the GMD, RMSE and FAC2 values for the median and average results using six members are close to the results for the single members. These results indicate that the LMVE ensemble method is applicable even when the ensemble size is only two, although the bias, uncertainty, correlation and precision dramatically decrease as the

ensemble size increases. The proposed ensemble method is very useful for properly estimating Cs-137 concentrations, even under a limited ensemble size.





### 4.4 Cs-137 spatial distribution

The above discussion indicates that the LMVE ensemble method can better estimate the Cs-137 distribution; the spatial distributions of Cs-137 concentrations that are integrated daily on 15 March 2011 are also shown (Figure 11). In the Fukushima prefecture, including the FDNPS, the results of Cs-137 simulated by each member are largely spread, so the

ensemble results, especially in the area far from the observation sites, are very important. Figure 5 suggests that the ensemble results are moderately correlated with the observations around the area where the distance from the observation site is approximately 20-30 km. Therefore, in the Fukushima prefecture which presents a complex terrain, parts of the results of Cs-137 in this area are still uncertain, even when using models with a 3 km horizontal grid, because part of the area is far (30 km away the coast of Fukushima, which is called Hama-dori in Japan), and the inner area is the location of many observation

sites (called Naka-dori in Japan). In contrast, although the difference in the simulated plumes among each member is very large over the Kantou region, the conclusion from Section 3.2 supports the results that the ensemble Cs-137 results are closer to the observations, with PCC>0.4, compared to the results of each member because in the Kantou region, most areas are within approximately 50 km of any observation site. However, some of the prefectures in the Kantou region do not have measurement sites, so in these prefectures, the ensemble results are still uncertain. This suggests that in the future, it should

be required to observe Cs-137 at every 20-30 km distance near the source region and every 50 km distance in other areas to properly estimate the best results of the Cs-137 spatial distribution.

### 5 Conclusions

The LMVE ensemble method is based on a classical idea but is still useful for estimating the best results using MMEs and

observations without requiring a large amount of computer resources for high-resolution models. This method was first applied to estimate Cs-137 distribution by Nakajima et al. (2017) and is extended in this study. The uniqueness of this approach compared with other MMEs for other species is based on the following: (1) the availability of observed Cs-137 concentrations near the surface at approximately 100 sites, providing dense coverage over eastern Japan; (2) the simplicity of identifying the emission source of Cs-137 associated with the point source of FDNPS; (3) the novelty of implementing the

MME approach with a high-resolution model over complex terrain in eastern Japan; and (4) the strong need to better estimate the Cs-137 distribution due to its inhalation exposure risk among residents in Japan. However, Nakajima et al. (2017) did not thoroughly discuss the availability of this method in depth, show the biases, uncertainties, precision and generalizability of this method under varying time-windows, space-windows and ensemble sizes. Radioactive Cs-137 was released from the FDNPS in March 2011, and many studies have investigated the distribution of Cs-137, but the proper

estimations of Cs-137 are not still adequate. Therefore, this study first extended the LMVE ensemble method to an ensemble size of six for simulating Cs-137, including two models, the WRF-CMAQ and NICAM models, and observations and then investigated their uncertainties to confirm their performances to generalize this method and attempt to give the best estimate


for the estimation of their inhalation impacts on humans, which is a companion study by Takagi et al. (to be submitted). The results of the ensemble members are also updated from Nakajima et al. (2017) by using a finer horizontal resolution (3 km grid) and by nudging an improved meteorological field provided by Sekiyama et al. (2017).

The proposed LMVE ensemble method provides the best results among the single members of the ensemble. This shows that

the MME-estimated Cs-137 concentrations at all available 101 sites have the lowest bias against the observations, with GMB=1.53; the lowest uncertainties, with RMSE=9.42 Bq m$^{-3}$; the highest correlation against the observations, with PCC=0.58; and the highest precision against the observations, with FAC2=54%. Moreover, the single-model members provided higher biases (GMB=1.20-4.29), higher uncertainties (RMSE=19.2-51.2 Bq m$^{-3}$), lower correlation coefficients (PCC=0.29-0.45) and lower precision (FAC2=10-29%). In the model grid excluding the observations, Cs-137 is estimated

by a spatial interpolation of variance in the formulation of the LMVE using the inverse distance weighting between the nearest two sites. For the test of this assumption, the available measurements are divided into two sets, learning and validation data, and thus, the test finds that the assumption for linear interpolation promises a moderate PCC value (>0.4) within a distance of 0.4°-0.5°, i.e., approximately 50 km. Extra sensitivity tests for several parameters, i.e., the site number and the weighting coefficients in the spatial interpolation, the time window in the LMVE and the  ensemble size, are

determined. As a result, the findings for the uncertainty in the proposed LMVE ensemble method are shown: (1) The LIP1 method (using two sites and IDW) is the simplest and provides better results than the other interpolations. (2) The time window in the LMVE ensemble method can be set to 1 hour. (3) A larger number of ensemble members (ensemble size) remarkably yielded better results, and more than two members had better results than each member alone. Therefore, the proposed LMVE ensemble method with a maximum ensemble size has the potential to provide the best estimate of the Cs-

137 distribution, even under a limited ensemble size (at least two).

It should be noted that, however, the LMVE ensemble method presents certain limitations. When Cs-137 simulated by all members is too underestimated compared to the observations, the variance in Cs-137 between the observations and the simulations, as defined in Eq. (3), must be too large and, thus, the weighted coefficient of the members, as defined in Eq. (5), becomes very small. Because the cross terms of the Cs-137 concentration and the weighted coefficient are small, the Cs-137

concentrations estimated by the ensemble must be underestimated. In contrast, when Cs-137 simulated by some members is overestimated compared to the observations, the weighted coefficient becomes very small. However, because the cross terms of the Cs-137 concentration and the weighted coefficient are not small, the Cs-137 concentrations estimated by the ensemble are overestimated.

In addition, the spatial interpolation used in this study does not obtain a moderate PCC value (>0.4) in the areas where the

distance from the observation sites exceeds 50 km. Therefore, the estimated results over the area with very sporadic site locations are very uncertain, especially for the inner areas of the Kantou region, e.g., Gunma and Tochigi prefectures. The assumption of the spatial interpolation using IDW is difficult to apply to broadly distributed materials, such as Cs-137 emitted from the FDNPS, which is spatially and temporally distributed very heterogeneously (Nakajima et al., 2017). It can be said that it is difficult to use any spatial interpolation, which basically assumes the spatially smoothness of the target's





concentrations based on the best estimation of the Cs-137 distribution. In the future, Cs-137 should be observed at distance intervals of 20-30 km near the source region including complex terrain and intervals of 50 km in other areas to properly estimate the best results of the Cs-137 spatial distribution.

This study only applies the LMVE ensemble method to radioactive Cs-137, but this method can be applied to atmospheric

pollutants, such as PM2.5. Recently, the large areal coverage of surface PM2.5 measurements is available in most countries (https://aqicn.org/map/world/). In addition, since the PM2.5 distribution does not vary abruptly, the LMVE ensemble method is easily applied for PM2.5 estimations. Therefore, this method will be applied in our future study to estimate the PM2.5 distribution for better air quality prediction.

## Data availability

The WRF-CMAQ and NICAM model results used to support this article can be obtained from the corresponding author by request (goto.daisuke@nies.go.jp). The observational data of Cs-137 is freely accessible at http://www.radiochem.org/paper/JN152/jn15201_Appendix_A_rev.pdf (accessed on August 2019).

## Author contribution

DG designed the experiments, conducted the ensemble calculation, and drafted the manuscript. YM and JU conducted model

simulations. TO, TTS and TN contributed to the discussion of the ensemble results. All authors contributed to write the manuscript and all of the discussion.

## Competing interests

The authors declare that they have no conflict of interest.

## Acknowledgements

We acknowledge the relevant researchers, Haruo Tsuruta and Yasuji Oura, for measuring the Cs-137 concentrations at the monitoring sites. Some of the authors were supported by the Environmental Research and Technology Development Fund (5-1501 and 1-1802) of the Environmental Restoration and Conservation Agency, Japan; the Ministry of the Environment, Japan; JSPS KAKANHI Grant Number JP17H04711; and the Japan Aerospace Exploration Agency (JAXA)/Earth Observation Research Center/GCOM-C.





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



Table 1. Brief model description and the design of the experiments

| Name | W-model (W1) | W-model (W2) | W-model (W3) | W-model (W4) | N-model (N1) | N-model (N2) |
|---|---|---|---|---|---|---|
| Dynamic core | WRF | WRF | WRF | WRF | NICAM | NICAM |
| Module | CMAQ4.6 | CMAQ4.6 | CMAQ4.6 | CMAQ4.6 | Modified SPRINTARS[1] | Modified SPRINTARS[1] |
| Horizontal grid size (km) | 3 | 3 | 3 | 3 | 3 | 3 |
| Number of layers (lowest height) | 34 (19 m) | 34 (19 m) | 34 (19 m) | 34 (19 m) | 40 (20 m) | 40 (20 m) |
| Meteorological fields (nudged)[2] | SE17 | MSM | SE17 | SE17 | SE17 | MSM |
| Wet deposition for Cs-137[3] | WSPEEDI | WSPEEDI | CMAQ | WSPEEDI | SPRINTARS | SPRINTARS |
| Emission scenario of Cs-137[4] | KA15 | KA15 | KA15 | TE12 | KA15 | KA15 |

[1] Modified SPRINTARS optimizes SPRINTARS (Takemura et al., 2005) for simulating Cs-137 particles by assuming a one-modal size distribution with a radius centre of 0.24 μm and high hygroscopicity, similar to sulfate (Nakajima et al., 2017).

[2] SE17 represents the meteorological fields calculated by NHM-LETKF in Sekiyama et al. (2017). MSM represents mesoscale objective analysis data (MANAL) from the Japan Meteorological Agency (JMA).

[3] WSPEEDI is a model for simulating the radioactive materials developed by JAEA (Terada et al., 2008).

[4] KA15 and TE12 represent Katata et al. (2015) and Terada et al. (2012), respectively.





Table 2. Test experiments in the ensemble method for linear interpolation (LIP)

| Name | Nearest | LIP1 | LIP2 | LIP3 | LIP4 |
|---|---|---|---|---|---|
| Number of grids | 1 | 2 | 2 | 3 | 3 |
| Method | Nearest neighbours | LIP | LIP | LIP | LIP |
| Weighting | No | Inverse distance | Inverse square-distance | Inverse distance | Inverse square-distance |
| Spatial seamlessness | No | Yes | Yes | Yes | Yes |

5  Table 3. Test experiments in the ensemble method using the selected sites

| Description | | CTL | SEN1 | SEN2 | SEN3 |
|---|---|---|---|---|---|
| Number of learning sites | | 101 | 77 | 45 | 23 |
| Number of validation sites | | 0 | 24 | 56 | 78 |
| Maximum distance between the learning and validation sites | | - | 0.125° | 0.25° | 0.5° |
| Colours in the used learning sites | Red | Yes | Yes | Yes | Yes |
| | Yellow | Yes | Yes | Yes | No |
| | Green | Yes | Yes | No | No |
| | Blue | Yes | No | No | No |

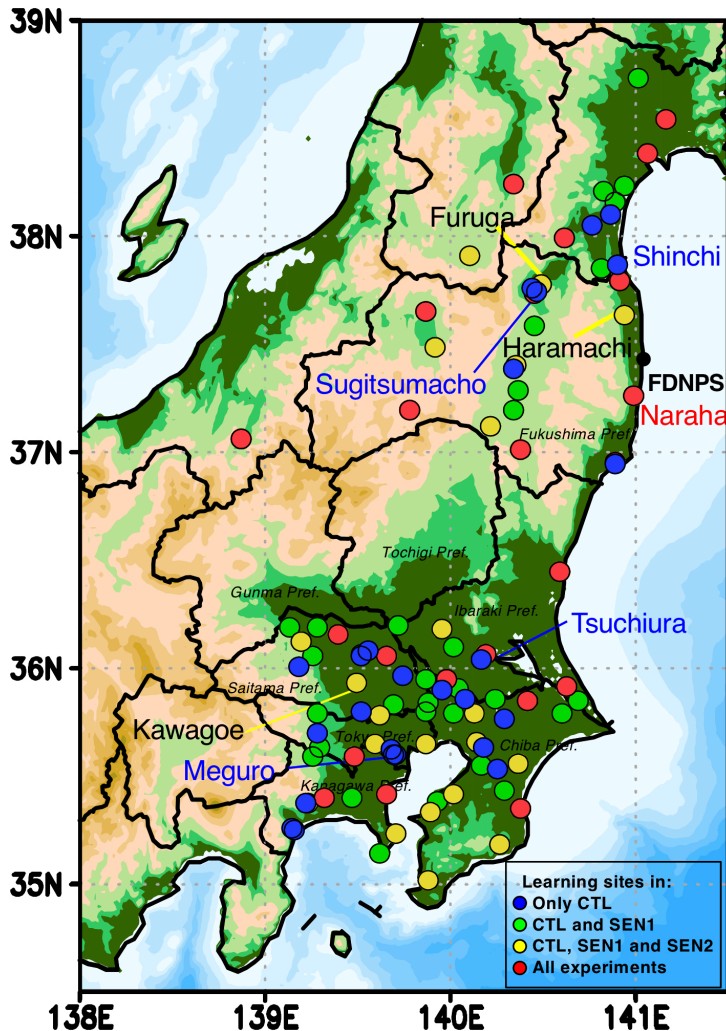

**Figure 1: Cs-137 observation sites used in this study. The closed black circle is the location of the FDNPS. The closed circles in red, yellow, green and blue are the locations of the learning data sites used in the standard experiment (CTL). The closed circle in yellow is the learning data location used for the ensemble in CTL, SEN1 and SEN2. The closed circle in green is the learning data location used for the ensemble in CTL and SEN1. The closed circle in blue is the learning data location used for the ensemble in the CTL only. The number of sites is 23 (red), 22 (yellow), 32 (green) and 24 (blue). The details are also explained in Table 3. The words in italic are the names of prefectures. The Kantou region includes seven prefectures; Tokyo, Kanagawa, Chiba, Saitama, Ibaraki, Tochigi and Gunma. The background map for the elevation is obtained from the ETOPO (Amante and Eakins, 2009).**

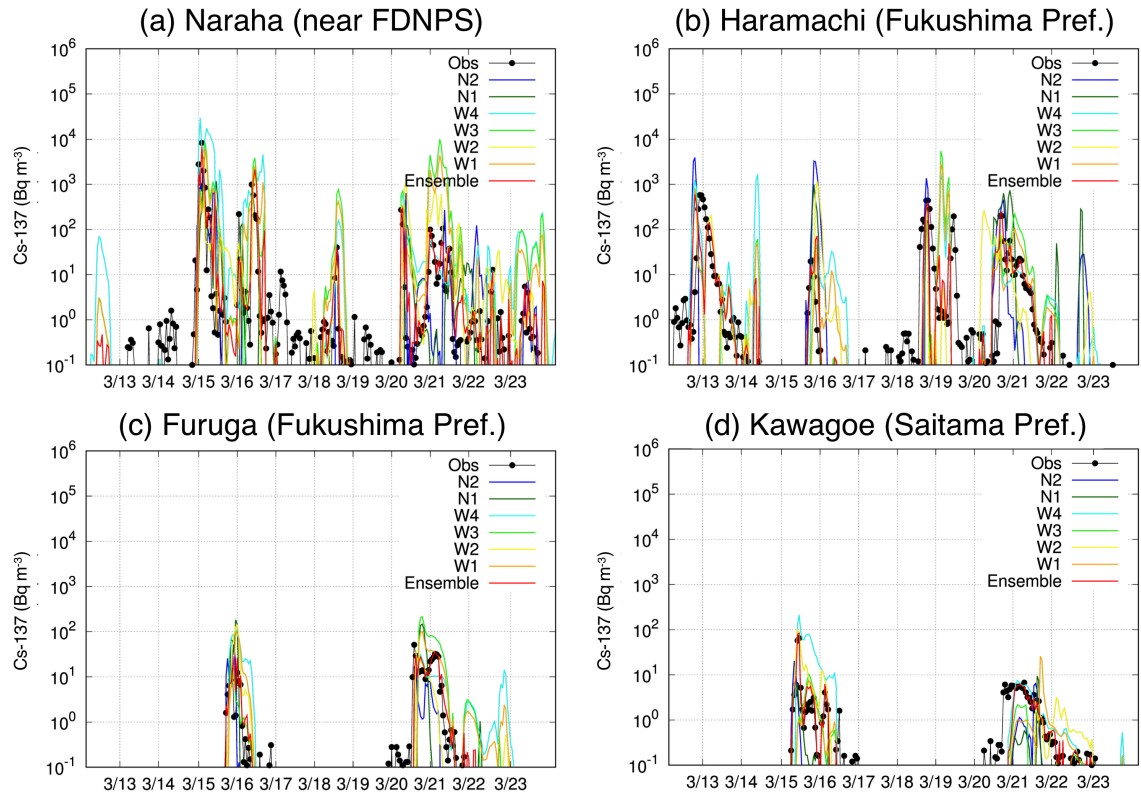

**Figure 2: Temporal variations in Cs-137 at the relevant sites (Naraha, Haramachi, Furuga and Kawagoe). The locations in brackets represent the names of prefectures. The results are shown for the observations ('Obs' in black), ensemble members (W1, W2, W3, W4, N1 and N2 in colours) and the ensemble model (red). The time is Japan Standard Time (JST).**



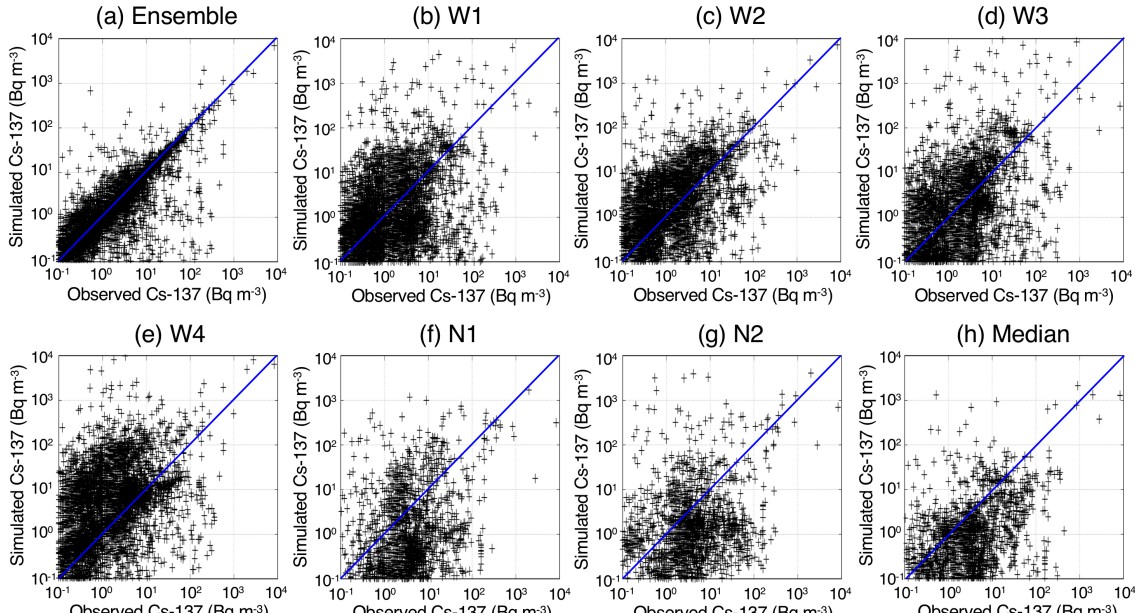

**Figure 3: Relationship between the simulated Cs-137 and the observed Cs-137 at all available sites using the (a) ensemble model with all models shown in (b)-(g), (b)-(g) original one-member model (W1, W2, W3, W4, N1 and N2) and (h) median model using all models shown in (b)-(g). The blue line is the 1:1 line.**



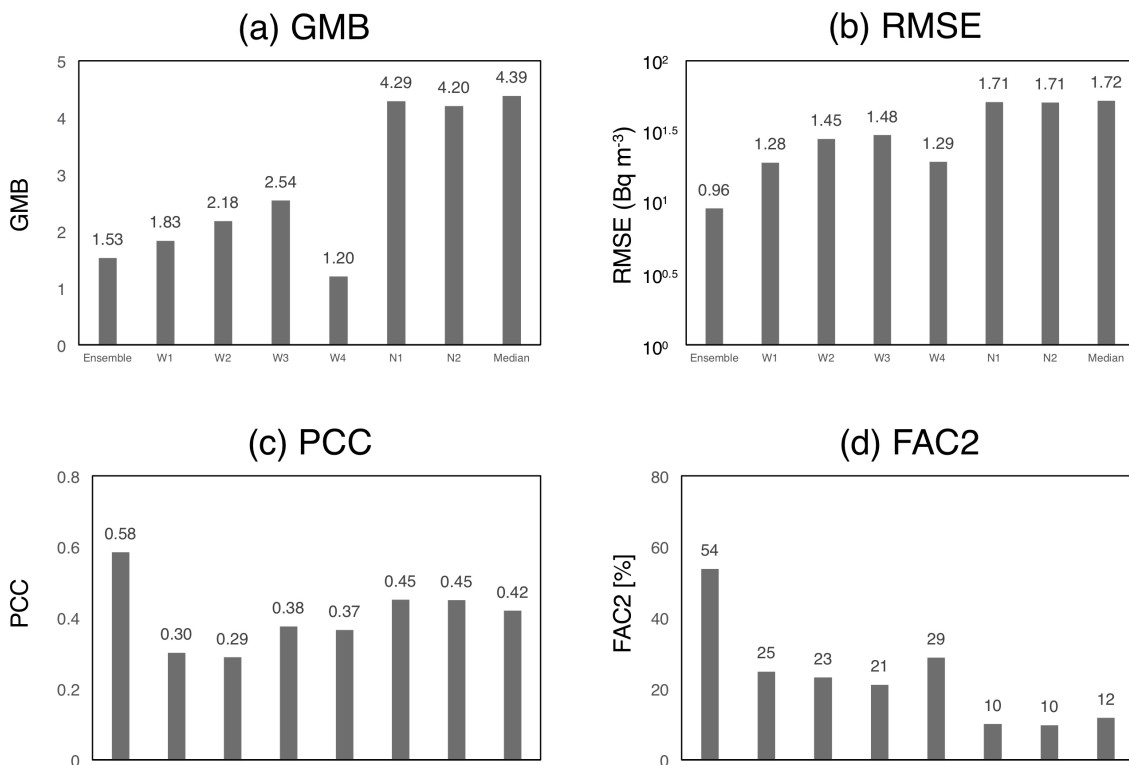

**Figure 4: Statistical metrics defined in Section 2.4 using the simulated Cs-137 and observed Cs-137 at the available 101 sites. The metrics shows the (a) geometric mean bias (GMB), (b) root-mean-square-error (RMSE), (c) Pearson correlation coefficient (PCC) and (d) fraction of data within a factor of two (FAC2). The results correspond to Figure 3.**





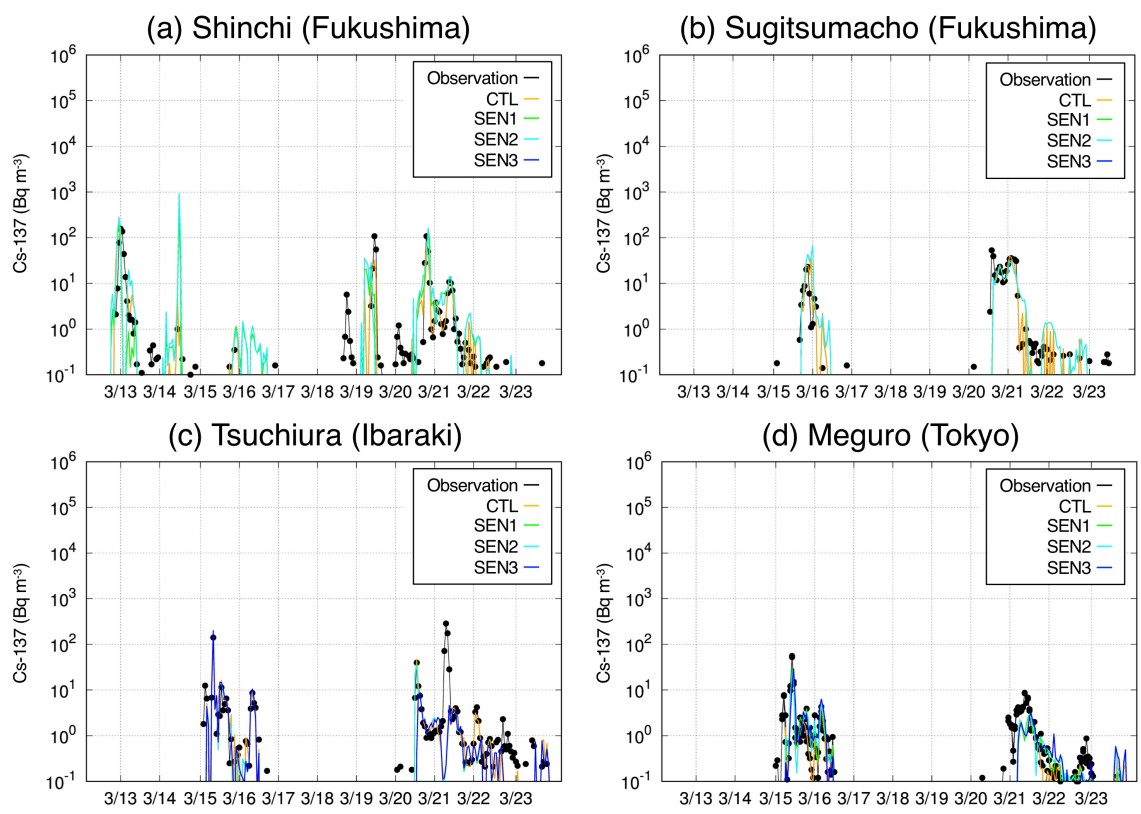

**Figure 5: Temporal variations in Cs-137 at the independent sites (not learning but validation sites) using the LMVE ensemble method for CTL, SEN1, SEN2 and SEN3. The time is JST.**





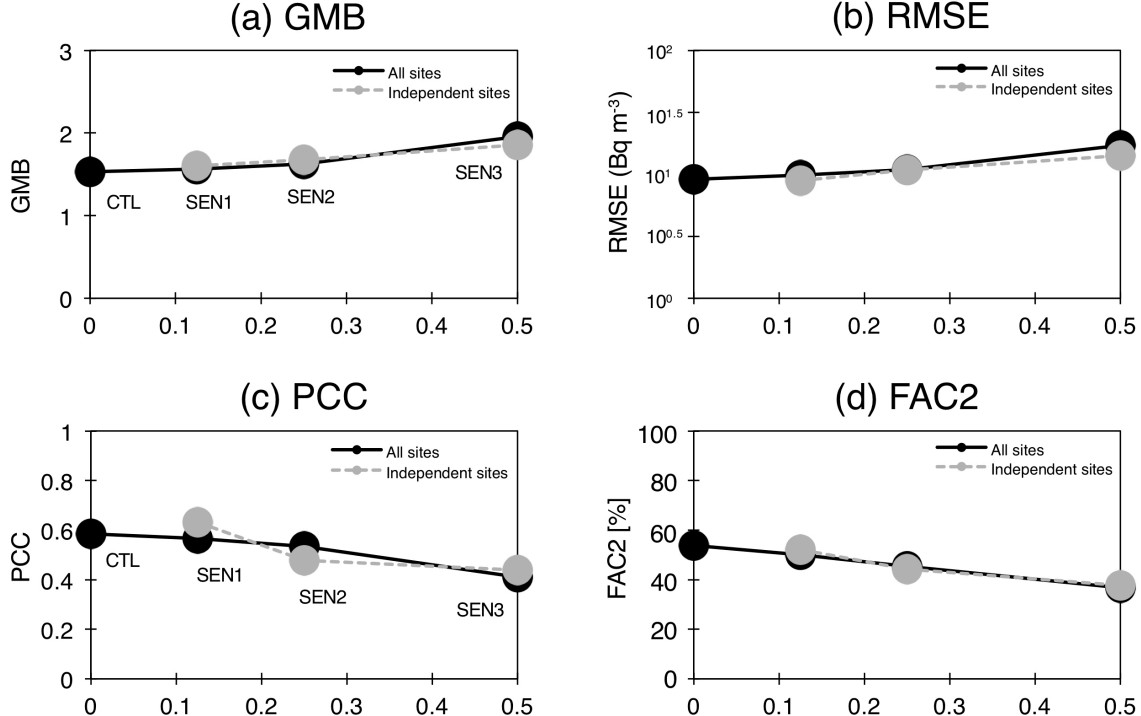

**Figure 6: Statistical metrics (GMB, RMSE, PCC and FAC2) at the available sites for CTL, SEN1, SEN2 and SEN3. The statistical metrics are calculated using all sites (in black) and the independent sites (in grey), which are not used in the LMVE ensemble method. The names of the experiments are shown in each panel. The X-axis represents a maximum distance between the learning and validation sites in units of degree.**





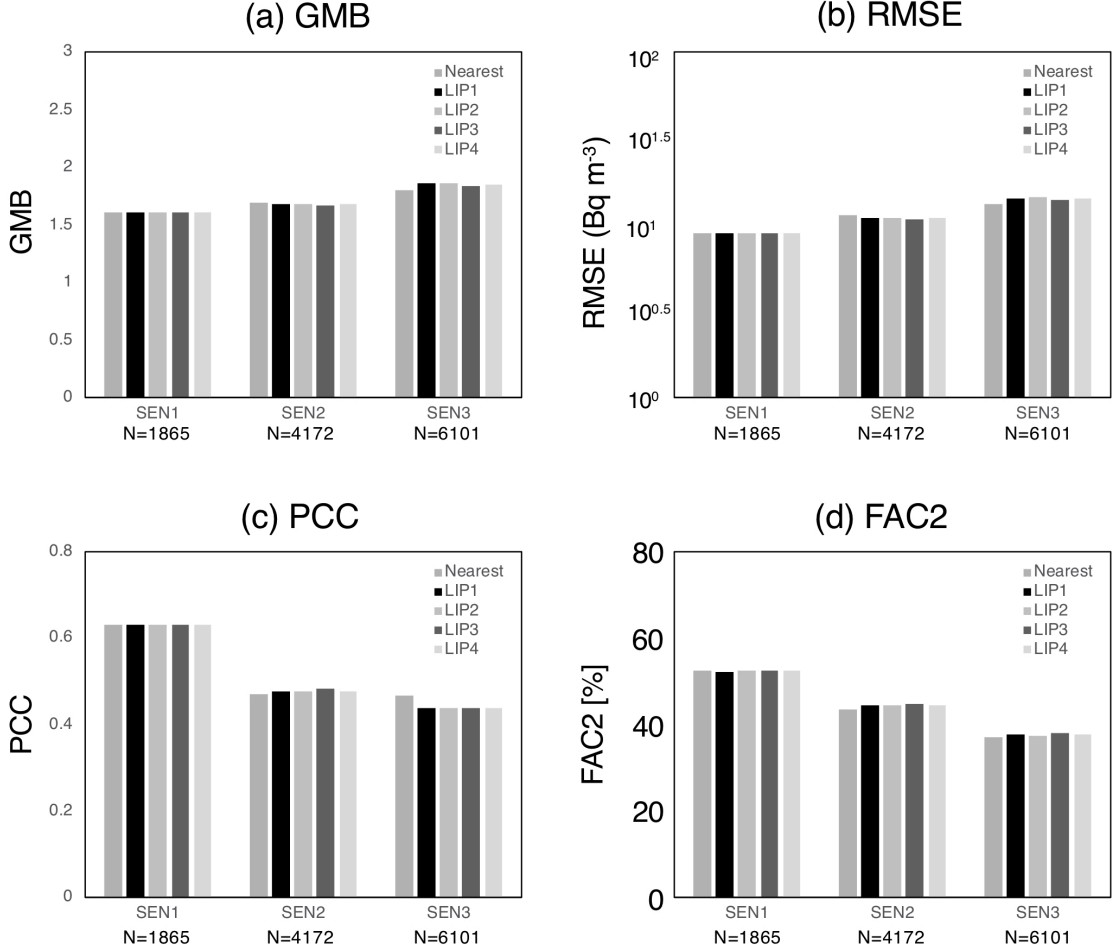

**Figure 7:** Statistical metrics (GMB, RMSE, PCC and FAC2) for the three sensitivity tests (SEN1, SEN2 and SEN3) as described in Table 3 using the five interpolation methods (nearest, LIP1, LIP2, LIP3 and LIP4) described in Table 2. The X-axis represents the sensitivity experiments with the sampling number (N).

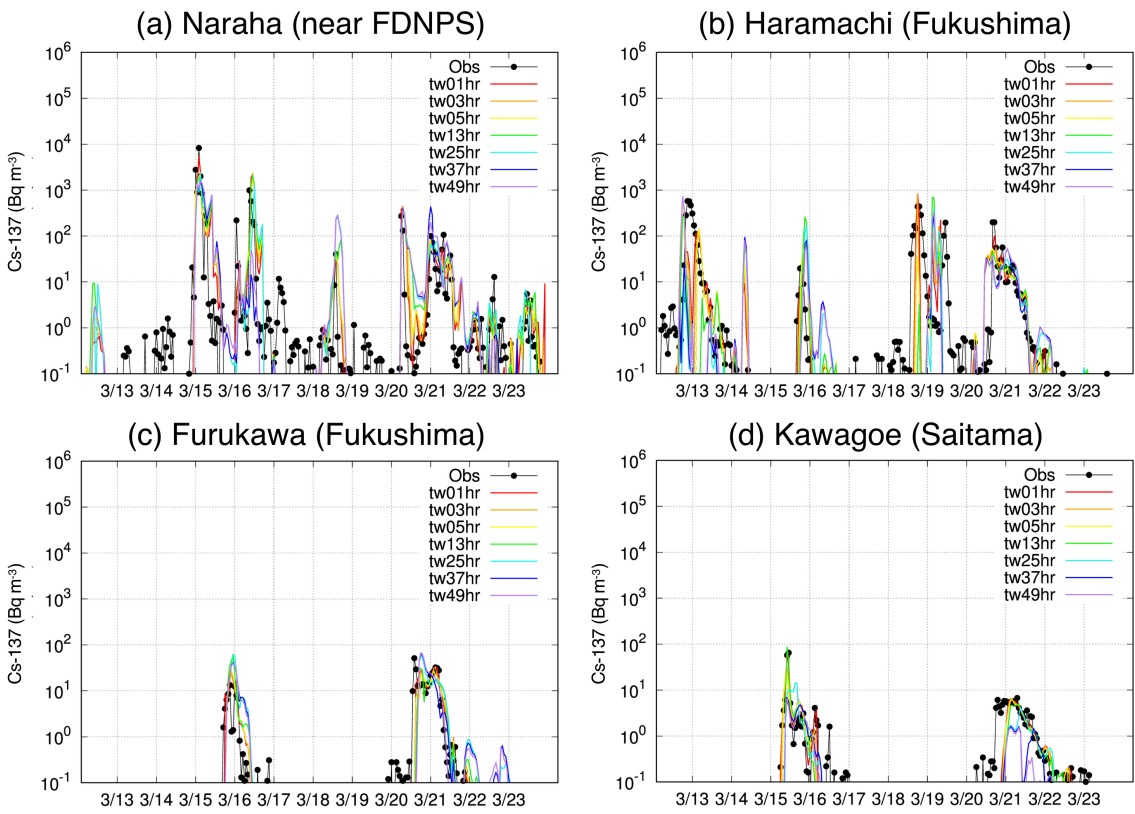

**Figure 8: Same as Figure 2 except for the use of the ensemble results with various time windows ranging from 1 hour (tw01hr) to 49 hours (tw49hr).**



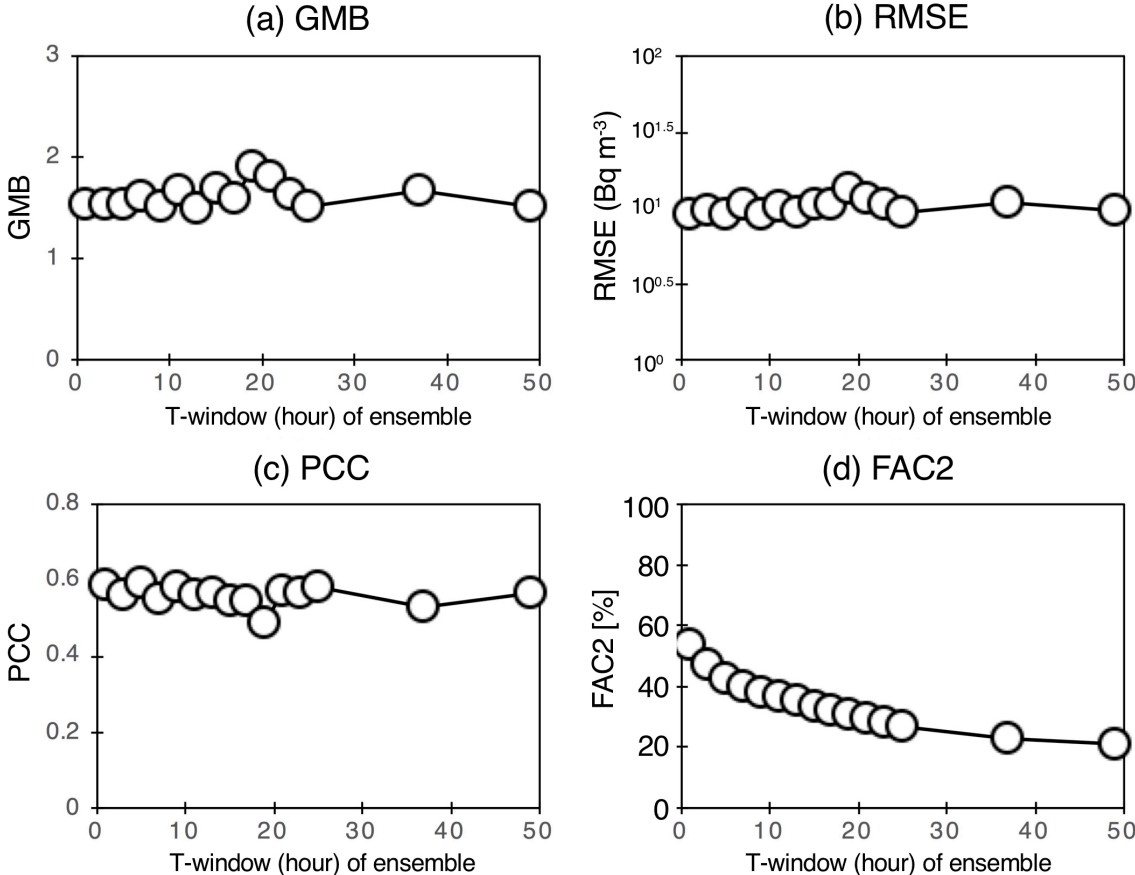

**Figure 9: Statistical metrics (GMB, RMSE, PCC and FAC2) at the available sites against various time windows (X-axis) ranging from 1 hour to 49 hours.**

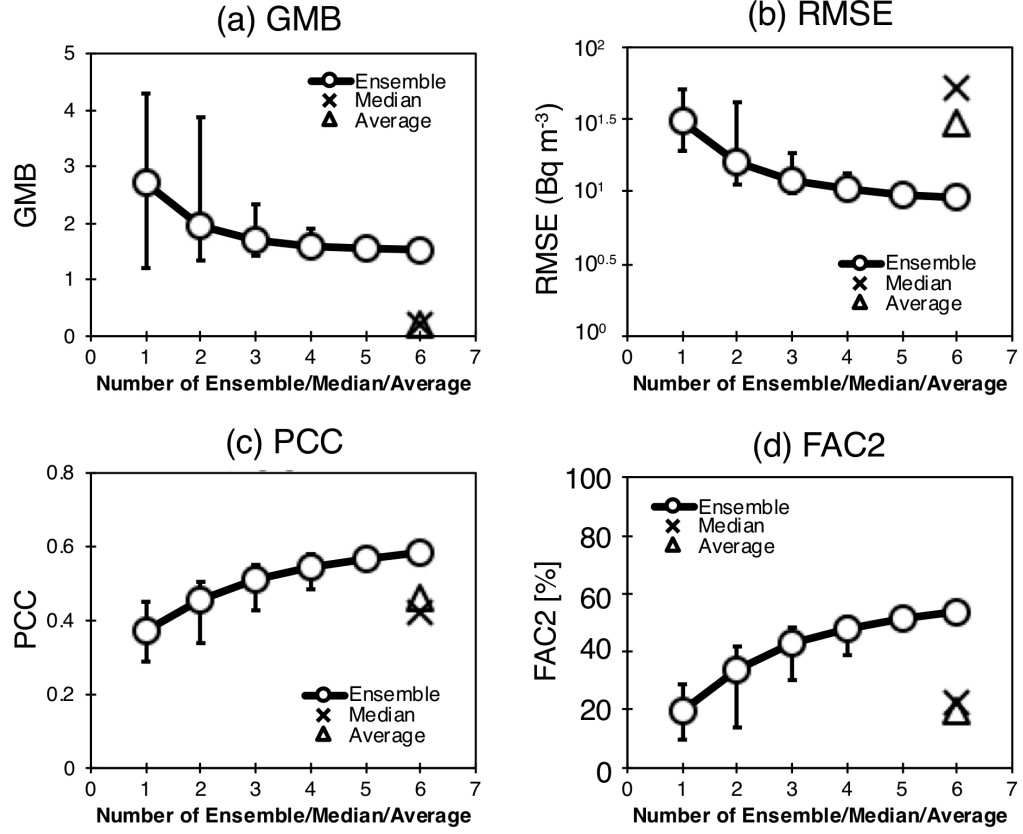

**Figure 10: Statistical metrics (GMB, RMSE, PCC and FAC2) at the available sites against the number of the ensemble members, the median and the average (X-axis).**

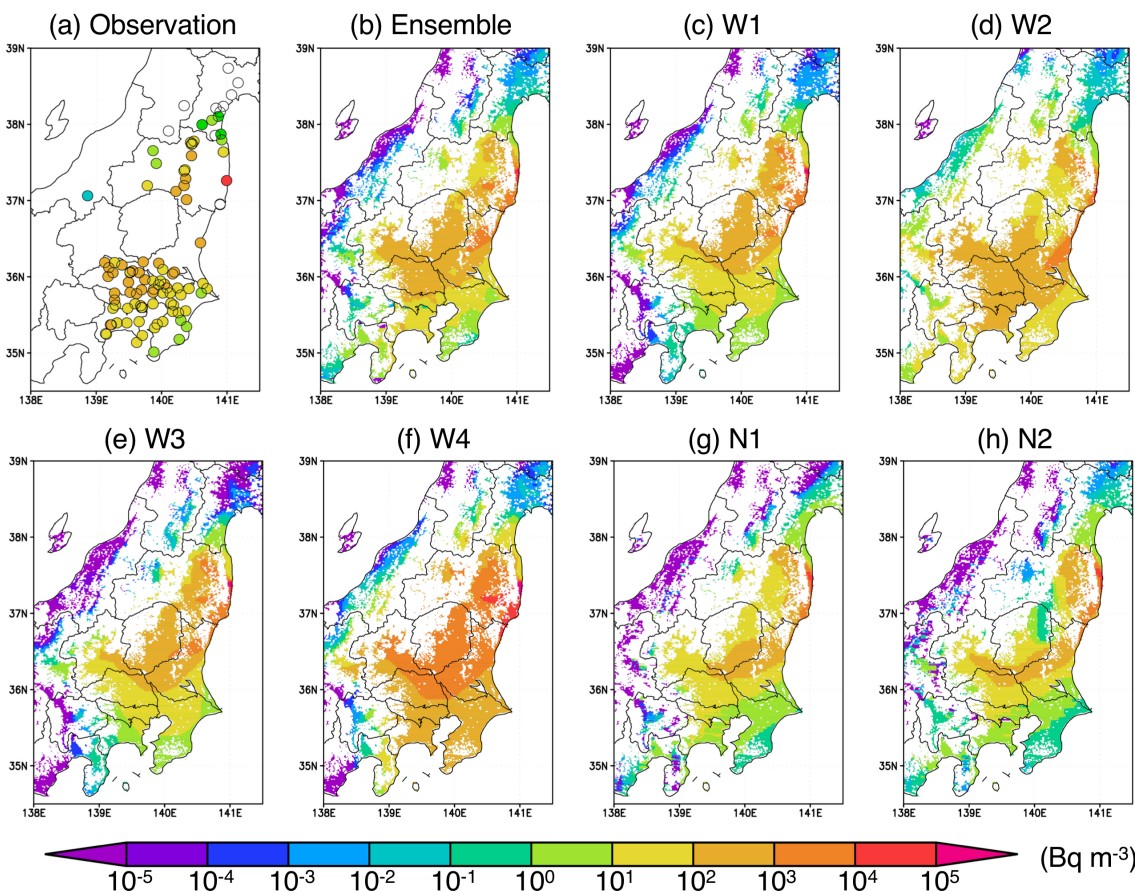

**Figure 11: Spatial distribution of the (a) observed and (b-h) simulated daily integrated Cs-137 concentrations on 15 March 2011 (JST).**