# Peer review of "Application of linear minimum variance estimation to the multimodel ensemble of atmospheric radioactive Cs-137 with observations"

_Atmospheric Chemistry and Physics, 2019_

## Referee Comment (RC1) · Anonymous Referee #1 · 8 Nov 2019

(Part A) [General comments] This paper presents a novel method that can significantly improve airborne Cs-137 predictions for ensembles of limited size and moderate performance. The authors use a linear minimum variance estimation and the data of approximately 100 sites covering eastern Japan, to combine CMAQ and NICAM model results for enhanced predictions. Numerical experiments with different data and various sensitivity studies have also been performed to demonstrate the behavior of this method, including the spatial interpolation, time window, and the ensemble size. With the optimism parameters, the proposed method shows very promising results and remarkable metrics. The ensemble simulation method could provide a more precise estimation of the nuclide dispersion, thus helping us better understand the impacts of inhalation

exposure on residents in Japan. The paper is definitely worth publishing. However, it is suggested that the authors address the following issues, to make the paper better presented. (Part B) [Specific comments] In section 2.2, the weight ai seems to be calculated for data from each monitoring site. However, Figure 11 indicates that the weights are applied to the whole calculation domain. In section 4.1, the author studied the interpolation method of variance, which may seem to be used for weight calculation beyond site positions. However, this explanation may not be easily found by the readers. It is suggested that the authors add some explanations on how to apply the weights to the whole domain in section 2.2. In section 2.3, it would be helpful if the authors added the lattices of representative sites to Figure 1. These lattices may help the reader to understand the representativeness of these sites. Besides, it is suggested that the authors give some explanations on how to choose the representative site in these lattices. For example, it is possible to present the data set of each experiment in an individual subplot of Figure 1. In section 3.2, Figure 5 shows that SEN3 does not reproduce the observations at Shinchi and Sugitsumacho. But there is a learning site (the red one in Figure 1) which is close to Shinchi, which should provide some information. Would the authors add some explanations for the phenomenon? In section 4.2 Are all the observations used by the ensemble methods in the sensitivity tests of ensemble size and time windows? In section 3.2, Figure 6, what's the difference between the knots on the all sites line (black line)? Are these knots the metrics calculated from a part of the all-site ensemble results (those predictions at the sites used by SEN1, SEN2, and SEN3)? In section 4.2, why the GMB, RMSE, PCC remain stable while the FAC2 drop apparently due to the weakening of the peak by using the longer time window. Is it possible to discuss the deposition predictions of the proposed method? It could be interesting to see whether the air concentration correction can improve the deposition prediction as well. (Part C) [Technical corrections] Some description can be simplified to be concise and clear. 1 Introduction P1 L9 "great efforts have been carried out to simulate atmospheric pollutants" could be better. P3 L10 "limiting" should be "limited" P3 L33 "the available results were increased via the use of six members"

[Figure]

could be "The available results were increased to six members" 2 Method P4 L20 "the basic experimental design in this study is common, such as in Morino et al." could be "the basic experimental design in this study is widely used in this field . . ." P6 L7 "discussed in section 4.2" should be "discussed in section 4.1" 3 Results P9 L12 "in figure 5(a) and 5(c)" should be "in figure 5(a) and 5(b)" 5 Conclusion P13 L26 "when Cs-137 simulated by some members is overestimated compared to the observations" could be " when Cs-137 is overestimated by some members"

---

## Referee Comment (RC2) · Anonymous Referee #2 · 15 Dec 2019

I find the paper of interest, overall well balanced and presented, although too coincise in some aspects that, I think, deserve more detail (see minor comments below).

I advice the editor to accept the paper for publication to ACP.

I invite the authors to deepen the discussion about the size of ensemble. as it is presented seems that the message is 'the more the better', while has been largely proven that is rarely the case (rarely in the sense that only when combining truly independent models the MME improves reliability of each single member). Please comment on that.

additional references (andf reference therein) for you to consider: Atmos. Chem. Phys., 9,9471–9489 Atmos. Chem. Phys., 13, 8315–8333, 2013 Atmos. Chem. Phys., 15,

2535–2544, 2015 Atmos. Chem. Phys., 16, 15629–15652, 2016 Atmos. Chem. Phys., 14,11791–11815

---

## Author Comment (AC1) · 20 Jan 2020

(Part A)
[General comments]
[C1-1] This paper presents a novel method that can significantly improve airborne Cs-137 predictions for ensembles of limited size and moderate performance. The authors use a linear minimum variance estimation and the data of approximately 100 sites covering eastern Japan, to combine CMAQ and NICAM model results for enhanced predictions. Numerical experiments with different data and various sensitivity studies have also been performed to demonstrate the behavior of this method, including the spatial interpolation, time window, and the ensemble size. With the optimism parameters, the proposed method shows very promising results and remarkable metrics. The ensemble simulation method could provide a more precise estimation of the nuclide dispersion, thus helping us better understand the impacts of inhalation exposure on residents in Japan. The paper is definitely worth publishing. However, it is suggested that the authors address the following issues, to make the paper better presented.

[A1-1] Thank you so much for reviewing our manuscript. We would like to address your pointed issues to improve our manuscript. Thanks to your comments, we found our mistakes in some results (Figures 5, 6, 7, 9a, 9b, 9c and 11b), which requires small modification of the sentences but do not change our main conclusion. Also, according to the suggestion by the reviewer #2, we have reconsidered the best ensemble size and modified section 4.3 including Figure 10. Details are mentioned in the point-by-point qualification.

(Part B)
[Specific comments]
[C1-2] In section 2.2, the weight ai seems to be calculated for data from each monitoring site. However, Figure 11 indicates that the weights are applied to the whole calculation domain. In section 4.1, the author studied the interpolation method of variance, which may seem to be used for weight calculation beyond site positions. However, this explanation may not be easily found by the readers. It is suggested that the authors add some explanations on how to apply the weights to the whole domain in section 2.2.

[A1-2] Thank you for your suggestion. Yes, as you mentioned, the weighted coefficient $a_i$ is calculated at each monitoring site. As mentioned in L32 in Page 5 in our original manuscript, the $a_i$ in grids where the monitoring site does not exist is calculated by interpolating two grids where the monitoring site exists. The interpolation method is also described in the same paragraph in section 2.2, but as you pointed these may not be easily found by the readers. This part in the latter of section 2.2 are revised by incorporating Eq. (7) for the spatial interpolation.

[C1-3] In section 2.3, it would be helpful if the authors added the lattices of representative sites to Figure 1. These lattices may help the reader to understand the representativeness of these sites. Besides, it is suggested that the authors give some explanations on how to choose the representative site in these lattices. For example, it is possible to present the data set of each experiment in an individual subplot of Figure 1.

[A1-3] Thank you for your suggestion. Surely, Figure 1 may include various information and be difficult to the representative sites used in the individual experiments (CTL, SEN1, SEN2 and SEN3). In the revised manuscript, we have plotted four panels in revised Figure 1, which can help the readers to find the relevant sites. In addition, we have added the following comments to the explanation how to choose the representative site in these lattices; "*The validation data are selected by choosing the sites with the largest number of observed samples as the representative site in the specific domain*" in section 2.3.

[C1-4] In section 3.2, Figure 5 shows that SEN3 does not reproduce the observations at Shinchi and Sugitsumacho. But there is a learning site (the red one in Figure 1) which is close to Shinchi, which should provide some information. Would the authors add some explanations for the phenomenon?

[A1-4] I appreciate your comments. Thanks to your comments, we found our mistake in this figure. We have revised the figure and confirm that the results are consistent to our expect, which they are closer to the observation near the learning sites. But the results of all the sensitivity tests (CTL, SEN1, SEN2 and SEN3) are sometimes far from the observations at Fukushima, although they are generally close to the observation. Related to this mistake, we have corrected Figure 5 and slightly modified the relevant comments on the manuscript. Please see the tracks in section 3.2 of the revised manuscript.

[C1-5] In section 4.2 Are all the observations used by the ensemble methods in the sensitivity tests of ensemble size and time windows?

[A1-5] Yes, we used all the observations. We have added this points to the end of section 2.3 "*In the sensitivity tests discussed in section 4, we used all observation sites*" and added "*101 sites*" to the caption of Figure 9 in the revised manuscript.

[C1-6] In section 3.2, Figure 6, what's the difference between the knots on the all sites line (black line)? Are these knots the metrics calculated from a part of the all-site ensemble results (those predictions at the sites used by SEN1, SEN2, and SEN3)?

[A1-6] This figure also includes our mistake and a part of the caption is not clear. We have corrected them. The black line shows the ensemble results using both learning and validation sites, whereas the grey line shows the ensemble results using only validation sites. Please see the revised Figure 6. After the corrections, the relationship between the two axes, i.e., distance vs PCC, is slightly changed. Although the slopes of the linear fitting line were 0.43 in the original manuscript, they are newly estimated to be 0.18 for the results using all sites and 0.36 for the results using only the validation sites. Therefore, according to the approximate line, the distance is calculated to be 1.03° when all sites are used and 0.68° when the independent sites are used to obtain moderate correlation (PCC>0.4); thus, the proposed interpolation is generally applicable for the best estimation within a distance of 0.7°-1.0°, i.e., at least 70 km (0.4°-0.5°, i.e., approximately 50 km in the original manuscript), from the observation site. These are written in the last paragraph in section 3.2 of the revised manuscript, the value shown in abstract, that shown in section 4.4 and that shown in section 5.

[C1-7] In section 4.2, why the GMB, RMSE, PCC remain stable while the FAC2 drop apparently due to the weakening of the peak by using the longer time window.

[A1-7] Thank you for your comments. We recheck the results and found very simple mistakes by using different values when plotted in the Figure. GMB, RMSE and PCC in Figure 9 in the original manuscript were totally wrong. I apologize for confusing you. We have corrected the figure as Figure 9, which clearly shows a strong dependence on the time window. As the time window is long, very sharp peaks of the Cs-137 cannot be resolved, so that the shorter time window is better. This point was already mentioned by Figure 8. So, in the revised manuscript Figure 8 and the revised Figure 9 are consistent with each other, as we expected. The main conclusion is remained, i.e., the time window can be set to the shortest time (1 hour). We have also modified the last sentence of Abstract.

[C1-8] Is it possible to discuss the deposition predictions of the proposed method? It could be interesting to see whether the air concentration correction can improve the deposition prediction as well.

[A1-8] Thanks you for your comments. Of course, the proposed method can be applied to the deposition predictions by using the simulated and observed deposition fluxes. However, the weighted coefficients derived from the air concentration would not be matched with those derived from the deposition flux, because the simulated air concentration and deposition flux are not strongly correlated with each other as suggested by previous model comparison studies like Table 3 in Kitayama et al. (2018). As you suggested, this perspective could be interesting for readers, so we have added these points to the revised manuscript in the last paragraph in section 5 as follows; "… *can be applied to Cs-137 deposition and …. However, the results obtained in this study are not directly used for estimation of the best estimate of the Cs-137 deposition because the simulated Cs-137 concentration and the simulated Cs-137 deposition flux are not generally correlated with each other, as suggested by previous model comparison studies (Kitayama et al., 2018).*"

(Part C)
[Technical corrections]
[C1-9] Some description can be simplified to be concise and clear.
1 Introduction
P1 L9 "great efforts have been carried out to simulate atmospheric pollutants" could be better.
P3 L10 "limiting" should be "limited"
P3 L33 "the available results were increased via the use of six members" could be "The available results were increased to six members"
2 Method
P4 L20 "the basic experimental design in this study is common, such as in Morino et al." could be "the basic experimental design in this study is widely used in this field . . ."
P6 L7 "discussed in section 4.2" should be "discussed in section 4.1"
3 Results
P9 L12 "in figure 5(a) and 5(c)" should be "in figure 5(a) and 5(b)"
5 Conclusion
P13 L26 "when Cs-137 simulated by some members is overestimated compared to the observations" could be " when Cs-137 is overestimated by some members"

[A1-9] We appreciate your corrections. All of points are reflected on the revised manuscript.

[revised manuscript text omitted]

(a) Learning sites at CTL experiment  (b) Learning sites at SEN1 experiment  (c) Learning sites at SEN2 experiment  (d) Learning sites at SEN3 experiment

[Figure]

**Figure 1: Cs-137 observation sites used as a learning site in (a) the standard (CTL), (b) SEN1, (c) SEN2 and (d) SEN3 experiments in this study. The closed black circle is the location of the FDNPS. The closed circles in red, yellow, green and blue are the locations of the learning data sites used in the  CTL experiment . The closed circle in yellow is the learning data location used for the ensemble in CTL, SEN1 and SEN2. The closed circle in green is the learning data location used for the ensemble in CTL and SEN1. The closed circle in blue is the learning data location used for the ensemble in the CTL only. The number of sites is 23 (red), 22 (yellow), 32 (green) and 24 (blue). The details are also explained in Table 3. The words in italics are**

the names of prefectures. The Kantou region includes seven prefectures: Tokyo, Kanagawa, Chiba, Saitama, Ibaraki, Tochigi and Gunma. The background map for the elevation is obtained from the ETOPO (Amante and Eakins, 2009).

[Figure]

5   Figure 2: Temporal variations in Cs-137 at the relevant sites (Naraha, Haramachi, Furuga and Kawagoe). The locations in brackets represent the names of prefectures. The results are shown for the observations ('Obs' in black), ensemble members (W1, W2, W3, W4, N1 and N2 in colours) and the ensemble model (red). The time is Japan Standard Time (JST).

[Figure]

**Figure 3: Relationship between the simulated Cs-137 and the observed Cs-137 at all available sites using the (a) ensemble model with all models shown in (b)-(g), (b)-(g) original one-member model (W1, W2, W3, W4, N1 and N2) and (h) median model using all models shown in (b)-(g). The blue line is the 1:1 line.**

[Figure]

**Figure 4: Statistical metrics defined in Section 2.4 using the simulated Cs-137 and observed Cs-137 at the available 101 sites. The metrics shows the (a) geometric mean bias (GMB), (b) root-mean-square-error (RMSE), (c) Pearson correlation coefficient (PCC) and (d) fraction of data within a factor of two (FAC2). The results correspond to Figure 3.**

[Figure]

[Figure]

**Figure 5: Temporal variations in Cs-137 at the independent sites (not learning but validation sites) using the LMVE ensemble method for CTL, SEN1, SEN2 and SEN3. The time is JST.**

[Figure]

[Figure]

**Figure 6: Statistical metrics (GMB, RMSE, PCC and FAC2) at the available sites for CTL, SEN1, SEN2 and SEN3. The statistical metrics are calculated using all sites (in black) and the independent sites (in grey), which are not used in the LMVE ensemble method. The names of the experiments are shown in each panel. The X-axis represents a maximum distance between the learning and validation sites in units of degree.**

[Figure]

[Figure]

**Figure 7: Statistical metrics (GMB, RMSE, PCC and FAC2) for the three sensitivity tests (SEN1, SEN2 and SEN3) as described in Table 3 using the five interpolation methods (nearest, LIP1, LIP2, LIP3 and LIP4) described in Table 2. The X-axis represents the sensitivity experiments with the sampling number (N).**

[Figure]

**Figure 8: Same as Figure 2 except for the use of the ensemble results with various time windows ranging from 1 hour (tw01hr) to 49 hours (tw49hr).**

[Figure]

[Figure]

**Figure 9: Statistical metrics (GMB, RMSE, PCC and FAC2) at the 101 available sites against various time windows (X-axis) ranging from 1 hour to 49 hours.**

[Figure]

[Figure]

**Figure 10: Statistical metrics (GMB, RMSE, PCC and FAC2) at the available sites against the number of the ensemble members, the median and the average (X-axis).** The black line indicates an average of the ensemble results for each number, whereas the dash line indicates the maximum and minimum results of the ensemble results for each number.

[Figure]

[Figure]

**Figure 11: Spatial distribution of the (a) observed and (b-h) simulated daily integrated Cs-137 concentrations on 15 March 2011 (JST).**

---

## Author Comment (AC2) · 20 Jan 2020

[C2-1] I find the paper of interest, overall well balanced and presented, although too coincise in some aspects that, I think, deserve more detail (see minor comments below).
I advice the editor to accept the paper for publication to ACP.

[A2-1] Thank you so much for reviewing our manuscript. We would like to address your pointed issues to improve our manuscript. Through the revision to answer questions from the reviewer #1, several figures were corrected, and the related comments were modified in the revised manuscript, but our main conclusion and methodology are not changed.

[C2-2] I invite the authors to deepen the discussion about the size of ensemble. as it is presented seems that the message is 'the more the better', while has been largely proven that is rarely the case (rarely in the sense that only when combining truly independent models the MME improves reliability of each single member). Please comment on that. additional references (andf reference therein) for you to consider: Atmos. Chem. Phys., 9,9471–9489 Atmos. Chem. Phys., 13, 8315–8333, 2013 Atmos. Chem. Phys., 15, 2535–2544, 2015 Atmos. Chem. Phys., 16, 15629–15652, 2016 Atmos. Chem. Phys., 14,11791–11815

[A2-2] Thank you for your suggestions. We consider your recommended literatures and rethink about the ensemble size. As Potempski and Galmarini (2009) pointed in the introduction, the following general question is very important; "Which criteria should be adopted to guarantee that the ensemble results will always be superior to those of any individual member?" To attempt to answer this question, previous studies (Solazzo et al., 2013; Kioutsioukis and Galmarini, 2014; Solazzo and Galmarini, 2015; Kioutsioukis et al.., 2016) introduced an analytical solution by considering weighting coefficients and selecting the informative members and investigated model ensemble results from model intercomparison projects such as AQMEII and HTAP. These results indicate that an effective ensemble size, which can be determined by the model members and the focusing parameters, is always smaller than the number of the ensemble members. Solazzo et al. (2013), for example, pointed out the effective number of the ensemble size is 4-6 from all 13 members in AQMEII participated models. By considering these references, we rewrited our discussion in section 4.3 (sensitivity of the ensemble size) and slightly modify the implicit idea that "the more the better".

Figure 10 in the original manuscript clearly shows that as increase the ensemble size, GMB becomes close to 1, RMSE decreases, and both PCC and FAC2 increase. This qualitative tendency is generally consistent to the previous studies of Figure 4 in Solazzo et al. (2013), Figure 3 in Pennell and Reichler (2001), Figures 1 and 11 in Kioutsioukis and Galmarini (2014) and Figure 3 in Solazzo and Galmarini (2015). Therefore, we can also say that "The addition of more models to the ensemble is not compensated by a linear increase in the overall information", as noted by Pennell and Reichler (2001) and Solazzo et al. (2013). In this study, "*the best ensemble results obtained using five members are close to those obtained using six members, with differences of +0.1% for GMB, +1.5% for PCC, +0.0% for RMSE and +0.2% for FAC2. By contrast, the best ensemble results obtained using four members are worse than those obtained using six members, with the differences of -1.6% for GMB, -0.9% for PCC, +0.8% for RMSE and -4.0% for FAC2. Therefore, we conclude that the minimum number in the LMVE ensemble in this study is five, but only when the members are effectively selected. When the members cannot be selected, the best results can be obtained by reducing the weighting coefficients of the members through the calculation of the LMVE method, since the difference in the statistical metrics between six- and five-member ensembles is very small."* These are incorporated to Section 4.3 in the revised manuscript.

As the reviewer pointed, the model member is not truly independent on the others and this is the usual case in the air quality models. Surely, the statistical metrics in median and average values in Figure 10 of the original manuscript are not good, probably because the ensemble members are not i.i.d. "independent and identically distributed around the true value" used in previous studies like Kioutsioukis and Galmarini (2014). This probably causes moderate correlation (NOT strong

correlation) between the ensemble and observation, i.e., PCC in the best results in this study is less than 0.7. Therefore, to obtain much closer to the observation, a new ensemble member, which should not be generally dependent on the current members, is required. However, it should be noted that even our proposed method provides a better result using too many ensemble members, as shown in Kioutsioukis and Galmarini (2014). This statement is obtained because the ensemble results become close to the average from the all members due to the central limit theorem. Therefore, we have added the following comments on the revised section 4.3; "*Even in the best estimate using selected five- or six-members, the PCC value is less than 0.7, which means the ensemble results are moderately (NOT strongly) correlated with the observations. Therefore, to obtain values much closer to the observations, a new ensemble member is required. As explained in section 3.1, when one of the members provides results close to the observations even at 1 hour, the ensemble results proposed in this study become closer to the observations*". In the revised manuscript (P.2 L5-7), we conclude that "*the important assumptions were ... the ensemble size; i.e., ... a larger ensemble number (
[revised manuscript text omitted]